# Photosynthetic Metabolism under Stressful Growth Conditions as a Bases for Crop Breeding and Yield Improvement

**DOI:** 10.3390/plants9010088

**Published:** 2020-01-10

**Authors:** Fermín Morales, María Ancín, Dorra Fakhet, Jon González-Torralba, Angie L. Gámez, Amaia Seminario, David Soba, Sinda Ben Mariem, Miguel Garriga, Iker Aranjuelo

**Affiliations:** 1Instituto de Agrobiotecnología (IdAB), Consejo Superior de Investigaciones Científicas (CSIC)-Gobierno de Navarra, Av. Pamplona 123, 31192 Mutilva, Spain; fmorales@eead.csic.es (F.M.); maria.ancin@unavarra.es (M.A.); fakhet.dorra@hotmail.fr (D.F.); angie-421@hotmail.com (A.L.G.); amaia.seminario@unavarra.es (A.S.); david.soba@unavarra.es (D.S.); sinda.ben@csic.es (S.B.M.); 2Dpto. Nutrición Vegetal, Estación Experimental de Aula Dei (EEAD), CSIC, Apdo. 13034, 50080 Zaragoza, Spain; 3Institute for Multidisciplinary Applied Biology, Dpto. Agronomía, Biotecnología y Alimentación, Universidad Pública de Navarra, Campus Arrosadia, 31006 Pamplona, Spain; jon.gonzalez@unavarra.es; 4Centro de Mejoramiento Genético y Fenómica Vegetal, Facultad de Ciencias Agrarias, Universidad de Talca, Talca 3460000, Chile; miguel.garriga@gmail.com

**Keywords:** climate change, crops, gas exchange, growth, photosynthesis, yield

## Abstract

Increased periods of water shortage and higher temperatures, together with a reduction in nutrient availability, have been proposed as major factors that negatively impact plant development. Photosynthetic CO_2_ assimilation is the basis of crop production for animal and human food, and for this reason, it has been selected as a primary target for crop phenotyping/breeding studies. Within this context, knowledge of the mechanisms involved in the response and acclimation of photosynthetic CO_2_ assimilation to multiple changing environmental conditions (including nutrients, water availability, and rising temperature) is a matter of great concern for the understanding of plant behavior under stress conditions, and for the development of new strategies and tools for enhancing plant growth in the future. The current review aims to analyze, from a multi-perspective approach (ranging across breeding, gas exchange, genomics, etc.) the impact of changing environmental conditions on the performance of the photosynthetic apparatus and, consequently, plant growth.

## 1. An Introduction to Climate Change and Crop Yield

The global population is forecast to increase by 2 billion and reach 9.8 billion people in 2050 [1]. This means that it is mandatory to raise crop productivity by 70% to meet the projected demand by the middle of the century [2]. Food quality should also be taken into account because it must provide all the essential nutrients to maintain human health. Alongside the increase in population, changes in the diet, particularly due to the higher meat consumption in some developing countries, may exacerbate the demand for feed crops.

Abiotic environmental stresses are considered as major limitations threatening worldwide food security and have a great impact on global crop production. It has been proven that the impact of these changes on natural systems and human health has already been harmful. It is reported that the global mean land and ocean surface temperatures have increased by 0.8 °C during the period 1888 to 2012 [3,4]. Further, although some degree of uncertainty exists in how the global surface temperature will rise, worldwide averaged surface temperature is foreseen to increase by 1.4 to 5.8 °C by 2100 [3]. In addition, extreme heat events have been detected since 1950, and it is assumed that such extreme events are going to occur often in the future [4,5]. In fact, heat waves are projected to be more intense and longer-lasting, while cold episodes are projected to decrease significantly. Such changes are projected to occur almost everywhere [6].

Concurrent with the elevation of ambient temperatures, increases in evaporation and reductions in precipitation rates are expected, while a growing inequality in the distribution of precipitation around the world will make water reserves increasingly scarce [4]. Uncertainty, even greater than that of temperature, is inherent in the Earth’s water cycle. Projected geographical distribution of rainfall foresees increases in the north of Europe, a large part of Asia, north of North America, north-west of South America and center of Africa, and decreases in the south of Europe, Africa, Australia, North America, South America, north of Africa and north and east of South America [4] affecting, if confirmed, water availability in different parts of the world in different ways, and therefore, food production. In other words, climate warming will lead to increased high-temperature periods, drought periods, and floods [4,7]. Such changes in environmental conditions will induce considerable disequilibrium in crop production.

Elevated temperature has been described to cause faster crop development and thus to reduce crop duration, which is mainly associated with lower yields [8]. It has been reported by Schlenker and Roberts [9] that under climate change scenarios with a temperature rise of less than 1 °C, crop productivity of maize and soybean will decrease by more than 50%. Adding to that, Hatfield and Prueger [10] reported that warmer temperatures mainly impacted the reproductive stage of maize development, and grain yield was significantly reduced by as much as 80%−90% with respect to normal temperature regimes. Besides, increasing temperatures adversely affect plant growth and development, which could affect wheat productivity negatively. For each degree rise in temperature, wheat production is estimated to reduce by 6% [11]. Water availability is another factor that largely determines yield. According to Lesk et al. [12], globally, it is estimated that losses of cereal production amounting to 1820 million kg have been caused by droughts during the past four decades. In wheat crops, water deficits can diminish production by at least 60% [13]. Daryanto et al. [14] reported 40% and 60% yield losses in maize and bean, respectively, with ≈35% reduction in water; however, they indicated that the yield losses varied as a function of the phenological stage affected by drought. In general, for cereals (maize, wheat, and rice), the later phase of grain filling is more susceptible to drought than the vegetative stage [14,15]. Overall, a trend towards yield reduction is being observed despite the existence of breeding programs aimed at developing new genotypes that are more efficient under limiting conditions, and this reflects the combined impact of all environmental factors on crop production worldwide. While the impacts of individual stress factors have been investigated during recent decades, the interactions between them and among them have received (comparatively) less attention [16]. In fact, when increased atmospheric CO_2_ concentration is studied as a single factor, crop production tends to increase, but under field conditions, various stress factors can occur simultaneously, such as water scarcity and high temperature, which could mitigate the positive effect of high CO_2_ on plant yields. Obviously, crop yields are constrained by the environment during the crop growth period. Therefore, climate models could be helpful tools to predict trends in crop productivity under future climate scenarios [17].

In this review, we first address the photosynthetic metabolism under stress conditions, especially high temperature, water, and nutrients scarcity. We then describe major methods that have been used for photosynthesis-based breeding under the scenario of climate change. Finally, we discuss perspectives for crop breeding programs.

## 2. Photosynthetic Metabolism under a Changing Environment

Exposure to environmental stress induces numerous physiological reactions in plants. Among the key physiological changes are alterations in photosynthetic rates and assimilate translocation [18], changes in water uptake and evapotranspiration [19,20], effects on nutrient uptake and translocation [21,22], modifications to antioxidant reactions [23,24] and programmed cell death [25], and altered gene expression and enzyme activity [26,27], all of which are the most frequently impacted processes under environmental stress conditions.

In general terms, stress-derived inhibitory effects on photosynthesis may be due to (i) limitations on CO_2_ diffusion factors and/or (ii) metabolic factors (Figure 1). There is considerable evidence that underlines stomatal closure as the main event in stress conditions [28,29,30,31]. The result of stomatal closure is a decrease in the sub-stomatal and chloroplast CO_2_ concentration (Ci and Cc, respectively), which produces a fall in the assimilation of CO_2_. On the other hand, when water stress is more severe, metabolic limitations occur [32]. Among the most widely considered of these are impacts on Calvin cycle enzyme activity (Rubisco, SBPase, etc.) and depleted availability of ATP and NADPH [33,34].

### 2.1. High Temperature

Plant development and growth processes are predicted to be negatively affected by high temperatures in a climate change scenario. Faced with such an extreme global warming situation, carbon uptake by vegetation is clearly a major pathway for mitigating climate change [35,36]. Nevertheless, CO_2_ availability to plants can vary depending on the environmental conditions, and its assimilation is dependent on numerous factors; stomatal closure, changes in the activity of the Calvin cycle, or alterations in the thylakoid structure, among others [37,38]. Exposure of plants to high-temperature conditions results in well-documented changes at the biophysical and biochemical levels, thus limiting photosynthetic activity [39].

The effects of extreme high temperatures on crops have been widely analyzed in several species; wheat [40], cowpea [41], soybean [42], and rice [43]. In the case of wheat, Asseng et al. [44] provided a detailed impact assessment of high temperatures on yield and how critical the adverse effects will be on the production of this crop. Although most of these studies focus on the flowering stage, many evaluations have also been carried out on the other stages of plant development [10,45,46]. For example, Wang et al. [47] found differences in photosynthesis, the transpiration rate, and stomatal conductance values between pre-anthesis high temperature acclimated wheat plants and those that had not experienced pre-anthesis heat exposure. As Hay and Walker [48] described, crop yield is affected by heat stress due to temperature controlling the rate of plant metabolic processes. In this regard, many responses to heat stress have been identified in plants: growth inhibition, a seed establishment disability, a higher transpiration slowdown, a continued water loss, or a decrease in crop quality [49]. A decline in chlorophyll content is also one of the effects observed in plants exposed to high-temperature conditions, as described in cucumber and wheat [50], and it is due to a decrease in chlorophyll biosynthesis or acceleration of its degradation resulting from the destruction of enzymes involved in their biosynthesis [51]. 

In the first instance, extreme temperatures affect CO_2_ assimilation at the stomatal level, i.e., closure of stomata under heat stress conditions, and impaired photosynthesis is a consequence. In fact, by controlling water vapor and CO_2_ diffusion, stomata largely contribute to both the rate and the temperature dependence of photosynthesis [37]. Several studies have analyzed plant behavior at the stomatal conductance level under heat-stressed situations, with most indicating inhibition of photosynthesis [49,52,53,54]. Yang et al. [55] also suggested that stomatal responses to high temperature are strongly influenced by water vapor pressure in maize plants. 

Another step in the CO_2_ diffusion pathway from the atmosphere to the site of carboxylation in chloroplasts is when CO_2_ molecules pass through leaf mesophyll, with mesophyll conductance (gm) being an important component that limits this diffusion [56]. These authors describe this parameter as dynamic, which could be related to differences between photosynthetic efficiency/capacity. Furthermore, gm is finite and measurable, and of similar magnitude as stomatal conductance [57,58] and it varies between species [59] and in response to environmental variables, such as temperature [60]. Some authors have confirmed that gm is important because it explains a 40% decrease in CO_2_ concentration among atmosphere and carboxylation sites [61]. The data available in the literature indicate that gm exhibits a different pattern depending on the duration and the type of stress the plants experience [56]. In the case of extremely high temperatures, several authors have observed different responses in gm, from a decrease to a considerable increase or even constant values [60,61,62,63,64]. The work of von Caemmerer and Evans [65] confirms that the gm response might be one of the most temperature-dependent parameters in the biochemical model of photosynthetic rates. In this way, carbonic anhydrase and aquaporins have been proposed as key factors provoking structural changes in cell wall thus providing adaptive and acclimation responses which could explain the diversity in the temperature response of gm, [56,66], an idea supported by work in rice and tobacco plants [59,60,67,68]. Crop productivity is determined by photosynthetic CO_2_ assimilation and respiration rates as both processes are temperature sensitive, although respiration is typically more sensitive than net photosynthesis to rising temperatures [69]. Temperature affects photosynthesis by influencing the electron transport capacity of the thylakoid membrane, together with its action on the kinetics of Rubisco and carboxylation efficiency [70]. As leaf temperature becomes higher, the photosynthetic rate increases, until decreasing after reaching optimum temperature, showing the effect of temperature on photosynthetic CO_2_ fixation, and CO_2_ release by photorespiration and mitochondrial respiration [71]. In C3 plants, Rubisco is the principal enzyme responsible for carbon assimilation, although it can also assimilate O_2_, which competes with CO_2_ for enzyme binding sites. As substrates of Rubisco, both O_2_ and CO_2_ play a key role in regulating the response of photosynthesis to heat stress [72], with the CO_2_ fixation rate of plants being affected by the CO_2_/O_2_ ratio. The increase in CO_2_ translates into an increase in the CO_2_/O_2_ ratio, which causes a decrease in photorespiration. However, elevated temperatures stimulate photorespiration due to greater solubility of O_2_ relative to CO_2_ [73], and decrease the specificity of Rubisco towards CO_2_ [74,75,76]. At the biochemical level, the heat lability of Rubisco activase could explain the limitation of photosynthetic activity under high temperatures, and this is because heat stress reduces the capacity of Rubisco activase to maintain the activation state of Rubisco [77,78]. Some studies have shown that Rubisco activity decreases when the temperature is raised as it does not allow a good balance between the activation/inactivation rates of Rubisco in cotton and tobacco plants [51,77]. However, in C4 maize plants, changes in the expression of a larger Rubisco subunit and limited recovery of the Rubisco activation state work as acclimation mechanisms [51]. Some studies have also shown that the photosynthetic parameter Vcmax (the maximum rate of Rubisco carboxylase activity) acts as a limiting factor on the rate of photosynthesis, having substantial roles in temperature acclimation [79], and it has been observed that generally, it increases as temperature rises to an optimal value and from there, it usually declines [80,81]. This decline in Vcmax under high temperatures is probably due to the dysfunction of Rubisco activase, resulting in a reduction in Rubisco activity. In addition, the principal element of the chloroplast electron transfer chain, PSII, may be damaged by elevated temperatures [82]. PSII is a heat-sensitive component of photosynthesis, and in addition to changes in the D1 protein and the plastoquinone pool, the influence of high temperature on photosynthetic components alters the energy distribution in PSII becoming oxidized. This could destabilize lipid–protein interactions, which could perturb the organization and function of PSII [49]. 

There is also a need that contributes to our understanding of the mechanisms by which plants adapt to heat stress. Because heat stress is considered an abiotic stress, several studies have focused on biochemical reactions, principally those related to hormones and primary and secondary metabolites, such as antioxidants [83]. At the molecular level, an alteration in the expression of genes leads to the synthesis of stress-related proteins, such as the activation of heat shock proteins (involved in signal transduction during heat stress), which are known to be an important adaptive strategy under this stress conditions [54]. 

Finally, not all plant species are able to cope with heat stress at the same level; plant sensitivity to high temperature varies according to the severity, duration, and developmental timing of the stress. However, all of them are forced to modulate their metabolism to prevent irreversible damage by reprogramming biological processes for stress adaptation [84]. For that reason, researchers are currently seeking to improve crop heat tolerance through the development of heat-tolerant genotypes via molecular breeding and genetic engineering [85,86], even though these techniques are still very expensive and time-consuming.

### 2.2. Water Stress

An early initial response of plants, when faced with water deficit, is the alteration of plant water relations with decreases in leaf water potential of leaf, turgor pressure, relative water content (RWC), and transpiration rate (E) [15]. Due to their close relationship with turgor pressure, which represents the driving force for cell expansion [87], cell growth and leaf expansion are the most sensitive processes to drought [88,89]. Studies in crops, such as wheat, barley, and rice, have reported growth as being limited by drought [90,91,92,93]. 

In this sense, the response of plants to maintaining the cell water content is to limit water loss through fast stomatal closure [87,88,94,95]. For this reason, CO_2_ diffusion from the atmosphere to the sub-stomatal cavity is reduced, which results in a lower stomatal conductance (g_s_), and which in turn is the main cause of the decreases in the photosynthetic rate (A) under water stress [94]. Several studies have demonstrated decreases in A and g_s_ in important crops under water deficit, such as wheat [92], rice [96], and grapevine [94]. Additionally, water use efficiency (WUE), which indicates the amount of biomass produced per unit of water used, is regulated mainly by g_s_ [97] and under moderate water stress conditions it generally increases due to a lower g_s_ and E, as shown in wheat [98,99] and bean [100]. Nevertheless, under severe water scarcity, the WUE can decrease [101]. In general, stomatal conductance is more strongly affected and is regulated by endogenous abscisic acid (ABA) levels that trigger a cascade of signaling pathways and, consequently, an efflux of anion and K^+^ from guard cells, which results in stomatal closure [95]. Studies in rice have demonstrated increases in ABA levels under lower water content, and the authors suggest that an increase in endogenous ABA is associated with mechanisms of drought escape [102]. 

The reduced CO_2_ diffusion through the stomata under water stress conditions is accompanied by a reduction in mesophyll conductance (*g_m_*) [56,88]. As already stated, *g_m_* is defined as the CO_2_ diffusion to the site of carboxylation in the chloroplast through the leaf mesophyll. In addition, *g_m_* is considered an important factor in the regulation of photosynthesis under different environmental situations [68] due to a rapid response under stress conditions that is even greater than *g_s_*. It exhibits a wide variability among species, functional groups, leaf forms, and developmental stages [56]. The reduced *g_m_* under water stress can be due to physical alterations in the intercellular spaces, biochemical changes, and/or membrane permeability [88], the latter apparently associated with expression and/or regulation of aquaporins [103]. Studies related to overexpression of the aquaporin NtAQP1 in tobacco plants result in a positive correlation with *g_m_* and photosynthesis [68]. Obviously, the CO_2_ concentration within the chloroplast (*C_c_*) is lower than in the sub-stomatal cavity (*Ci*), and this difference increases under water stress conditions [88]. 

Plants have developed interlinked strategies to overcome stress, and the biochemical responses are detected after a reduction in CO_2_ availability in the mesophyll [103]; among these, osmotic adjustment, osmoprotection, antioxidation, and scavenging defense systems have been reported as being important for drought tolerance [15]. Osmotic adjustment is considered an important mechanism to allow the maintenance of water uptake and cell turgor under stress conditions [88]. It consists of the overproduction of compatible solutes and ions, non-toxic in high concentrations, including soluble sugars, amino acids, organic acids, potassium ions, etc. [15]. Compounds such as proline, glutamine, and glycine-betaine can prevent protein denaturation and maintain them in native conformation at intermediate water content [104]. For instance, the enhancement of proline levels has been observed in maize, alfalfa, and wheat, and it has been associated with stress tolerance [105].

Water stress conditions also stimulate the production of reactive oxygen species (ROS), such as peroxide (H_2_O_2_), superoxide (O_2_^−^), and singlet oxygen (^1^O_2_), that cause membrane rupture and thereby affect photosynthesis [87,95]. However, in contrast to photosynthesis, plant respiration remains fairly constant during water stress, although mitochondrial electron transport is affected mainly through the alternative oxidase pathway, which maintains electron transport and prevents the formation of ROS [103]. The plant growth regulator, γ-aminobutyric acid, as well as free amino acids and sugars also play a role in scavenging ROS [15]. Other studies have shown an increase in the activity of scavenging enzymes, such as the enhancement of peroxidase activity under water deficit in durum wheat [106].

The plant response to water stress involves a complex of genes and processes regulated by interlinked genetic and biochemical mechanisms [87]. Changes in leaf biochemistry can result from the downregulation of photosynthetic metabolism machinery in response to a low supply of carbon substrate under prolonged water stress. In this way, several enzymes are downregulated or de-activated at low CO_2_ concentrations [88]. A decrease in the maximum velocity of carboxylation (Vcmax) indicates inhibition of Rubisco activity [103], as shown in plants of *Medicago truncatula* [100]. Other enzymes also decrease their activity, such as nitrate reductase [88]. In contrast, the production of antioxidant enzymes and stress-related proteins is upregulated under water stress [87]. 

In addition, water stress can activate cell-signaling pathways as an efficient response to environmental changes. The responses are triggered by secondary signal metabolites that include hormones (ABA, ethylene, cytokinins (CK)), ROS, and second messengers (sugars) [87,88]. Some findings indicate crosstalk among the ABA and CK hormones and the stress-signaling pathway. Within this pathway, redox signals act as regulatory agents where changes in the redox state regulate the expression of different genes related to photosynthesis. Furthermore, soluble sugars can be associated with hormones as part of the signaling network, and they tend to increase under water-scarcity conditions where they modify gene expression and proteomic patterns. Receptors and sensing proteins localized in membranes have a role in various signaling pathways [88]. 

Finally, as yield losses become more common under increasingly frequent and intense drought periods, the search is on for effective adaptation strategies to the effects of climate change on agriculture. It is, therefore, important to understand the responses and mechanisms by which plants can increase their adaptability to these adverse conditions. The approach should integrate photosynthesis responses, stomatal closure, osmotic adjustment, plant water relations, ROS scavenging and defense, and signaling and gene expression.

### 2.3. Impoverished Soils

Soil conditions determine leaf metabolism, and particularly photosynthetic metabolism [107]. Among those soil characteristics, it should be remarked the soil volume able to be explored by roots related to its compactness and soil physicochemical properties (nutrient and toxic elements composition and its capacity to hold and release water and nutrients). Soil nutrient availability and plant requirements vary greatly, depending on the nutrient type. Among the macronutrients, N (in the chemical forms of NO_3_^−^ and NH_4_^+^) is one of the elements that plants need to absorb from the soil. N-unstressed plants have between 2% and 5% dry weight (DW) range of concentrations found in different plant tissues, crop developments, and species [108]. Soils contain large amounts of total P, from which only the chemical forms H_2_PO_4_^−^ and HPO_4_^−2^ are used by roots. Growth is not limited by P when the range in the plant tissues is between 0.3 and 0.5% DW [108,109]. Soil K content (mostly inorganic) is in the range of 0.3% to 3%. Sandy soils can be problematic for growth, because of the high mobility of K in such soils, and the high demand that plants have for this mineral. Potassium is required by plants in amounts as high as 1–2% DW [108], making it the most concentrated mono-charged cation in the cells. Among the micronutrients, Fe, Mn, Cu, and Zn need to be highlighted. Soils with Fe contents of up to 3.2% are common [110]. However, in calcareous and alkaline soils, Fe occurs as oxides and hydroxides with very low solubility. At these high pHs, the Fe atoms free in the soil are at a concentration of 10^−10^ M, while plant requirements are ca. 10^−7^ M [111], causing Fe deficiency. Fe deficiency appears when Fe in the plant tissues drops below 50–150 μg g^−1^ DW [108]. Soils are generally quite rich in Mn, but only the cation in the Mn^+2^ redox state is readily available to plants for root uptake. For different species and environmental variables, the threshold values for Mn deficiency are 10–20 μg g^−1^ DW [108]. In soils that have not been contaminated with Cu, soils have contents in the range of 2 to 40 mg per kg. Plant roots absorb Cu as the divalent cation Cu^+2^ in well-oxygenated soils, or as Cu^+^ under flooding or in poorly oxygenated soils. Leaf Cu deficiency usually appears below 3–5 μg g^−1^ DW [108]. Soils non-contaminated by Zn have 10–80 mg of Zn per kilogram of soil. Plants absorb Zn as the divalent cation Zn^2+^, although the chemical form ZnOH^+^ is possibly also absorbed in high pH soils [108]. The threshold for leaf Zn deficiency is 15–20 μg of Zn per gram DW [108].

In agricultural areas, plants require favorable soil compaction, optimal soil water content, and nutrients, because crops control growth and yield as a function of the edaphic conditions and the environment. Commonly, photosynthesis responds to those stresses slower than does growth. In compacted soils, under drought or in soils poor in nutrients, photosynthesis is impaired because of different factors [107]. In P-deficient leaves and under drought (the latter only in some species), chlorophyll (Chl) remains fairly constant. Under deficiency of N and K, or of micronutrients, such as Mn, Fe, Zn, and Cu, however, Chl diminishes. Leaves with low content of Chl may or may not have low photosynthetic rates; only large decreases in Chl will have consequences in light gathering. In leaves affected by Fe deficiency, a curvilinear function is found between Chl and light absorption, linear at very low values of Chl and with small increases in light gathering at the highest Chl concentrations [112]. 

A low stomatal aperture limits photosynthesis under a variety of stress factors. Thus, it is commonly reported in response to drought and widely documented in compacted soil-grown plants or in those soils inducing the deficiencies of the macronutrients K and P, and the micronutrients Zn and Cu [107]. When drawing conclusions about stomatal limitations to photosynthesis, based simply on low stomatal conductance or low transpiration rates, one must be wary. It is true that a low stomatal conductance will lead to low photosynthetic rates, but it is also true that low rates of photosynthesis, independent of the causes, will lead to elevated CO_2_ levels within the leaf mesophyll close to the sub-stomatal chamber, which will cause partial stomatal closure. In any of the two situations mentioned above, stomata close, but the stomata are only a part of the resistance that the leaf poses to the diffusion of CO_2_ from the atmosphere to the carboxylation sites in the chloroplast. Once the CO_2_ molecules cross the sub-stomatal cavity, they need to travel across the different layers of leaf cells, encountering different barriers, to reach the chloroplast Rubisco enzyme. Under drought and limiting N and Mn, the mesophyll conductance to CO_2_ limits the photosynthetic rates [107]. An impaired photochemistry clearly limits CO_2_ fixation in leaves affected by Fe deficiency. Furthermore, this was documented from time to time with K deficiency [107]. In N- and K-deficient leaves, C fixation is limited by the cell biochemistry (Rubisco carboxylation activity, biochemical reactions within the Calvin-Benson cycle, and carbohydrate anabolism and transport), while it is not frequent in response to low water availability, soil compaction, or scarce amounts of micronutrients [107]. In plants grown in environmentally-controlled growth chambers and in the field with limiting Fe, leaf light-gathering, PSII photochemical reactions, and Rubisco carboxylation co-limit photosynthesis, downregulating all of them coordinately [113,114]. Under some circumstances, such as plants deficient in K or micronutrients, there has been evidence of altered chloroplast ultrastructure and function. Further investigation is required to elucidate whether such changes are actually damage or might be alterations that evidence adaptation to the stress conditions (see 107 for a detailed discussion).

Plants in the field interact with many microorganisms that reach the leaves via the atmosphere and the roots via the soil. At the soil level, microbes (such as symbiotic ecto- and endomycorrhizal fungi, N-fixing bacteria, and mutualistic microbes) are able to promote plant growth. Microbes act, moreover, as sinks for a part of the carbohydrates synthesized by the plant, enhancing, as a result, photosynthesis in the plants with which they interact. Soil compaction has been shown to impair American elm CO_2_ fixation in non-inoculated plants, whereas in symbiosis with ectomycorrhizal fungi, physiology, and, in consequence, growth was hardly affected [115]. Thus, symbiosis with ectomycorrhizal fungi can be advantageous in forest areas where human activities, fire, or other causes have compacted the soil [115]. In line with this, the symbiosis of ectomycorrhizal fungi with *Populus cathayana* improved electron transport and CO_2_ fixation rates, more markedly under low than under high water availability [116].

## 3. Photosynthesis as a Strategy to Improve Crop Yield

As mentioned before, the production of sufficient food to meet increasing population demands while maintaining environmental sustainability is one of the biggest challenges of the twenty-first century [1,2]. This has to be achieved despite increasingly variable weather patterns that are associated with global climate change.

Crop biomass production and yield are mainly derived from the cumulative CO_2_ assimilation rate during the growing season [117]. There is increasing evidence that to achieve a quantum boost to cereal crop yield potential, a major improvement in photosynthetic capacity and/or efficiency will be required [118]. Reynolds et al. [118] highlight the fact that, together with traits like optimizing partitioning to grain yield, the increase in photosynthetic capacity and efficiency (radiation use efficiency, RUE) can be a targeted approach to develop new germplasm better adapted to stressful growth conditions.

There is also evidence that historic gains in wheat yield potential have been associated with increased photosynthesis. Furthermore, basic research in photosynthesis has confirmed that substantial improvements are theoretically possible [119]. 

According to Passioura’s identity [120], grain yield (GY) is determined by: GY = LI × RUE × HI,
where LI = light interception by the crop; RUE = radiation use efficiency, and HI = the harvest index (ratio of harvestable versus total aerial biomass) (Figure 2). Since the Green Revolution, yield potential has continued to increase mainly due to improvements in HI. Despite a hypothetical limit to HI of 0.62 in wheat, there has been no systematic progress since the early 1990s from values of 0.50–0.55. Austin [121] predicted a maximum theoretical HI of 0.62 in winter wheat based on an extrapolation from the mean value (0.49) observed for the four most modern winter wheat cultivars characterized by Austin et al. [122] in the United Kingdom. These data indicate that the HI of two major food crops, wheat, and rice, is now approaching a plateau, and further increases in yield will necessitate an increase in productive biomass and, therefore, an increase in photosynthesis. The limited capacity to increase HI highlights RUE as a factor that needs to be further incorporated into breeding programs. Theoretical calculations reveal that wheat yield potential can be improved by up to 50% through the genetic improvement of RUE. 

RUE is largely determined in cereal crops by the interception of light necessary for photosynthesis (especially after canopy closure) and the phenological stage. Thus, the angle adjustment of the leaf and canopy architecture has an important role in the photosynthesis and productivity of these crops [118,123,124]. Modern varieties of cereal crops (e.g., wheat, rice) have more erect leaves [124,125], facilitating the penetration of light into the lower layers of the canopy. It has been suggested that an ideal plant ideotype in cereals is one that has an increased leaf angle from the top to the basal layers of the canopy [126,127,128]. Cultivars with upright leaves have greater light interception capacity that translates into significant increases in photosynthesis, biomass production, and yield in major crops, such as corn [129,130,131], rice [132,133,134], and wheat [119,135].

The process of senescence in plants is genetically controlled and determined by environmental conditions. During senescence, the concentration of leaf chlorophyll decreases, and thus, as these decreases reach a given threshold, the leaf photosynthetic capacity decreases. Plants with delayed senescence or stay-green usually have a longer period of grain filling, greater accumulation of assimilated CO_2_ throughout the season, and higher yields [136,137]. Some studies have reported a close relationship between functional stay-green and tolerance to drought and heat stress [137,138,139,140,141,142,143]. It has been proposed that chloroplast ultrastructure regeneration and better performance of the antioxidant system are responsible for the functional “stay-green” phenotype [144,145,146]. Thus, functional stay-green could be useful to design crops suitable for stressful environmental conditions.

## 4. Photosynthesis-Based Breeding under the Scenario of Global Climatic Change: Marker-Assisted Selection, Genomic Selection, and Genetic Engineering

Photosynthesis is a quantitative trait that involves multiple genes, regulatory mechanisms, and different metabolic pathways and plant structures working together [148]. Several studies have identified possible traits to be addressed to increase the efficiency of CO_2_ assimilation in crops, either through traditional breeding or using modern techniques of synthetic biology and genetic engineering [117,149,150,151,152,153] (Figure 3 and Figure 4).

Photosynthesis, and traits related to plant performance in general under abiotic stress, is complex and quantitative in nature [154,155]. Quantitative trait expression is controlled by many quantitative trait loci (QTLs), most of them with a small effect on the trait [156]. However, there is a significant natural variation in traits associated with photosynthesis in the available germplasm of major crops [117,153] that could be used for breeding purposes. In breeding programs, assessment of the genotypes is usually carried out under multiple environmental conditions to estimate the genotype × environment interactions necessary for selecting stable and high-performance phenotypes [157]. High throughput phenotyping (HTP) is a good example, being a successful combination of plant science, engineering, and computation for identifying and assessing plant traits that are key breading targets for crop improvements under stressful environments. Among the HTP techniques, remote sensing is currently the most widely used approach to evaluate these traits [158] (Figure 3). Remote sensing techniques can be summarized in five categories: (i) RGB cameras, (ii) spectral reflectance, (iii) thermal imaging, (iv) fluorescence imaging, and (v) active sensors (LiDAR and Radar). The advantage of HTP is that it can continuously provide a wide variety of crop information at different spatial scales. On the other hand, due to the enormous volume and variety of imaging and remote sensing data generated, one of the main limiting factors is the management and interpretation of all this information. Potent analysis and statistical tools are required, and improved software will be necessary to generate suitable data modeling that incorporates genotypic, phenotypic, and environmental variability so that this phenotyping bottleneck can be overcome and better crop breeding achieved for the stressed conditions projected for the future.

An excellent introduction to QTL mapping and marker-assisted selection (MAS) for plant breeding has already been provided by Singh and Singh [159]. Indeed, QTLs for photosynthesis-related traits could be useful for MAS in breeding programs to obtain genotypes with improved CO_2_ assimilation and WUE. Several studies of QTL mapping of traits associated with CO_2_ assimilation have been conducted in major crops. As was previously discussed, g_s_, canopy structure, and functional stay-green are among the most addressed traits studied to improve plant photosynthesis in crops. The relationship between QTLs and gs has been established in rice [160,161,162], wheat [163,164,165,166], barley [167,168], and sorghum [169], although a more in-depth mapping is required to enable the identification of genes directly related to the anatomy and functioning of the stomata. Changes in leaf angle and plant architecture are features of interest in cereal breeding programs [170,171]. Several QTLs related to leaf angle have been identified in maize and rice ([172] and references therein), bread wheat [173,174,175], and durum wheat [176], as well as genes associated with changes in leaf angle in maize, rice, and sorghum [172]. On the other hand, the association between QTLs and stay-green has been reported in wheat [137,143,177,178], maize [140,179,180,181,182], barley [141], sorghum [183,184,185,186], and rice [187]. In the same way, the association between chlorophyll content in the flag leaf and the magnitude of the net assimilation of CO_2_ has been reported, and QTLs for chlorophyll content and photosynthesis have been identified in rice [188], wheat [163,189], and barley [167,190]. These QTLs could be useful tools in MAS systems for improving photosynthesis under stressful conditions in these species. Although crop QTLs mentioned here are related to g_s_, canopy structure, and stay-green due the nature of the CO_2_ assimilation process, QTLs associated with stress tolerance, such as productivity traits (grain yield and yield components) and physiological traits (e.g., stem reserve mobilization or water-soluble carbohydrates), are to some extent, also associated with photosynthesis. 

In spite of the many studies and QTLs mapped in major crops, only a few have been used in practice for selection purposes in crop breeding programs, and those have been mostly restricted to traits with simple inheritance, such as monogenic or oligogenic inheritance including resistance to diseases and pests [191,192,193]. Among other factors, Jiang [194] stated that the low impact of MAS on crop breeding is due to (i) not all markers being applicable across populations due to lack of marker polymorphism or reliable marker-trait association, (ii) imprecise estimates of QTL locations and effects; (iii) a large number of breeding programs not being equipped with adequate facilities and conditions for large-scale adoption of MAS, (iv) MAS methods are not designed for large scale use in practical breeding programs, and (v) higher startup expenses and labor costs. Because of these limitations, and the development of next-generation sequencing (NGS) technologies that have allowed sequencing of genomes with high efficiency and decreasing costs, interest in the use of genomic selection (GS) tools has increased [195,196,197]. Genomic selection uses genome-wide markers (e.g., single nucleotide polymorphisms (SNPs)) and statistical models to predict the genomic breeding value of individuals in a breeding population [198,199,200]. Some examples of GS in plant breeding of maize, barley, wheat, rice, and oats for productivity and morphological traits, and disease resistance, are reviewed by Barabaschi et al. [201]. Nowadays, reference genomes for several important crop species (e.g., rice, wheat, barley, maize, and soybean) are available in public databases (http://www.gramene.org/; http://www.plantgdb.org/), and there are NGS-based platforms for genome-wide high-throughput genotyping for crop species [195,197]. Combining these with the accumulated information on QTL mapping, genome-wide association (GWAS) studies, and gene expression will all contribute to the consolidation of phenotype-associated genomic regions for crop breeding of quantitative traits [196]. 

Although GS is a promising approach to select parents in the early stages of the breeding process [195], the slow progress in high-throughput field phenotyping (HTFP) is limiting its implementation in breeding programs [157,202,203]. Well-designed phenotyping across multiple environments may provide more accurate estimates of QTL/gene locations and the effects necessary for efficient genomic selection [194]. High-throughput field phenotyping platforms capable of simultaneous assessment of traits of interest in dozens or hundreds of genotypes are currently implemented by big companies or big public institutions, such as the Australian Plant Phenomics Facility, the European Plant Phenotyping Network, and the United States Department of Agriculture (USDA), but these kinds of platforms are very expensive. Thus it is necessary to develop low-cost and easily manipulated phenotyping tools for their wide adoption in breeding programs [203] to reduce the gap between genomics and phenomics.

Advances in genetic engineering and synthetic biology make it possible to manipulate genes associated with photosynthesis with the aim of improving primary plant production. Several genes of interest for the improvement of photosynthesis and crop yield have been identified and manipulated to modify stomatal conductance, generate better ideotypes, reduce losses from photorespiration and respiration, enhance Rubisco efficiency, and increase sink strength and photoassimilate partitioning into the sink. These studies have been the subject of several reviews [149,150,152,153,204,205]. Despite this, little success has been achieved through genetic modification in improving leaf photosynthesis and WUE [204,205]. Flexas [204] listed some factors that contribute to this limited success: (i) improving gs results in a limited increase in photosynthesis and it is often at the expense of reducing WUE; (ii) there is scarce knowledge of the basis of mesophyll conductance; (iii) improving Rubisco efficiency has failed, and (iv) improving the amount and activity of key Calvin cycle enzymes has been achieved and results in improved photosynthesis. However, these studies do not report the effects on WUE. According to the same author, instead of single-gene/single-trait approaches, multi-gene approaches are necessary to improve photosynthesis and WUE, including those affecting traits like Rubisco, Calvin cycle enzymes, carbohydrate transport, stomatal and mesophyll conductance, and also those related to plant architecture. Perhaps the most ambitious project in this area is the introduction of C_4_ traits into rice funded by the Bill and Melinda Gates Foundation, which is intended to increase photosynthetic efficiency by 50%, improve nitrogen use efficiency, and double water use efficiency in rice (https://c4rice.com/). Some recent reviews highlight the basis of this work and its advances [206,207,208]. 

Despite the broad possibilities of modern molecular and synthetic biology tools for engineering crop photosynthesis, mining of the natural variation in existing germplasm through NGS, QTL, and SNP analyses, together with deeper and more practical phenotype screening, are currently the most feasible approaches for improving plant photosynthesis [150,204].

## 5. Perspectives

Regardless of the enhancer effect of high CO_2_ on plant net photosynthesis and productivity, predictions of climate change also involve rises in average temperatures of 2.6 °C and 4.8 °C by 2065 and 2100, respectively, and greater climate variability with more frequent periods of drought and heatwaves [209]. It is well established that both high temperature and drought have detrimental effects on photosynthesis and plant production [210,211]. When simulated climate change scenarios include elevated atmospheric CO_2_ concentration alone, plant responses are generally increased in photosynthesis, growth, and yield. However, growth and production may decrease under the future higher atmospheric CO_2_ concentrations when plants are also subjected to elevated temperature or insufficient water. In fact, significant decreases in yields of rice and wheat have been reported under elevated CO_2_ (500 ppm) due to an increase in canopy temperature of 1.5–2.0 °C [212]. 

Crop production must double by 2050 to meet the predicted production demands of the global population, however, achieving this goal will be a significant challenge for plant breeders. Traditional plant breeding is based on phenotypic selection and subsequent progeny testing, commonly followed by re-selection, which can be a very slow process and often requires time-consuming and costly phenotyping. The understanding of physiologic and molecular mechanisms involved in the response of photosynthetic machinery and plant development under changing environmental conditions could lead to the development of new strategies and tools for enhancing stress tolerance. There is increasing evidence that to achieve a quantum boost to crop yield potential, a major improvement in photosynthetic capacity and/or efficiency will be required. There is also evidence that historic gains in wheat yield potential have been associated with increased photosynthesis. Thus, it is recommended for crop breeding programs to continue working on the improvement of the efficiency of plant CO_2_ assimilation (through a better CO_2_ stomatal and/or mesophyll conductance, higher capacity for photosynthetic electron transport and CO_2_ fixation, and optimized sugar transport and use) and at the same time on the enhancement of the water (WUE) and nutrient use efficiency in preparation for the environmental conditions predicted for the future. Further research is also recommended on plants with delayed senescence (stay-green) since this situation usually maintains the plant photosynthetically active for a longer time leading to an extended period of grain filling.

## Figures and Tables

**Figure 1 plants-09-00088-f001:**
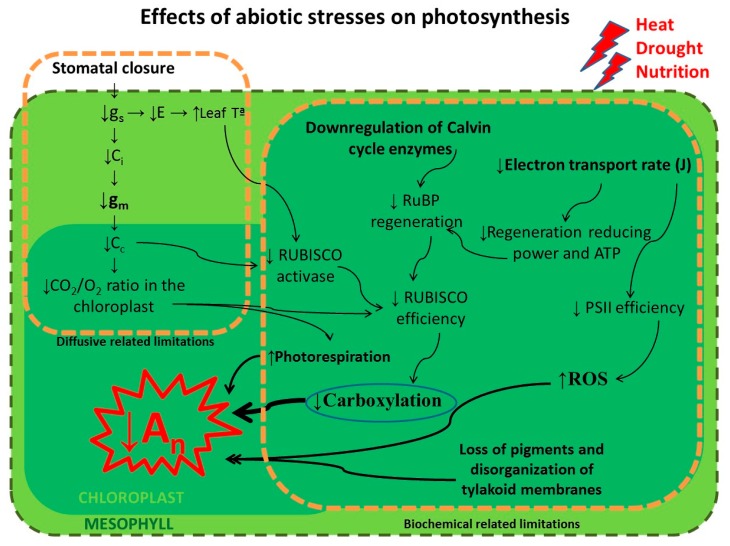
Photosynthetic metabolism under a changing environment. A multi-scale model representing the impact of key stress factors (high temperature, drought, and low soil fertilization levels) on photosynthetic performance associated with limitations to CO_2_ diffusion factors and/or metabolic factors. Stomatal closure diminishes sub-stomatal and chloroplast CO_2_ concentration (Ci and Cc, respectively) with a consequent reduction in photosynthetic rates (An). Under severe stress conditions, limitations to the activity of Calvin cycle enzymes (Rubisco, SBPase, etc.) and the photosynthetic electron transport rate (J), and the appearance of reactive oxygen species (ROS), among others, are observed.

**Figure 2 plants-09-00088-f002:**
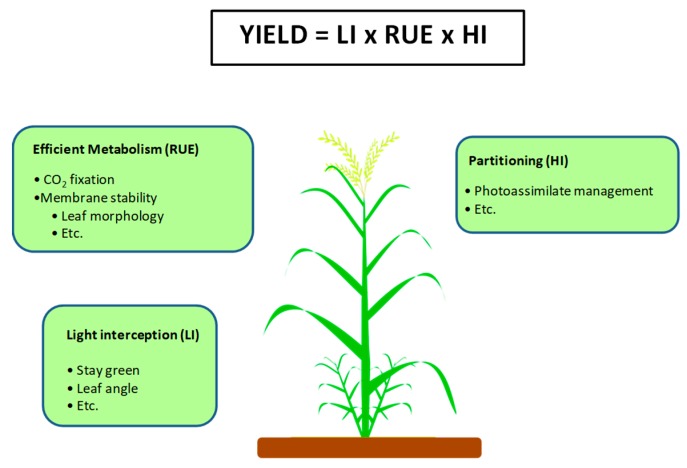
Photosynthesis as a strategy to improve crop yield. Grain yield improvement model developed by Passioura [120]. LI refers to light interception by the crop; RUE refers to radiation use efficiency, and HI refers to harvest index. Adapted from [147].

**Figure 3 plants-09-00088-f003:**
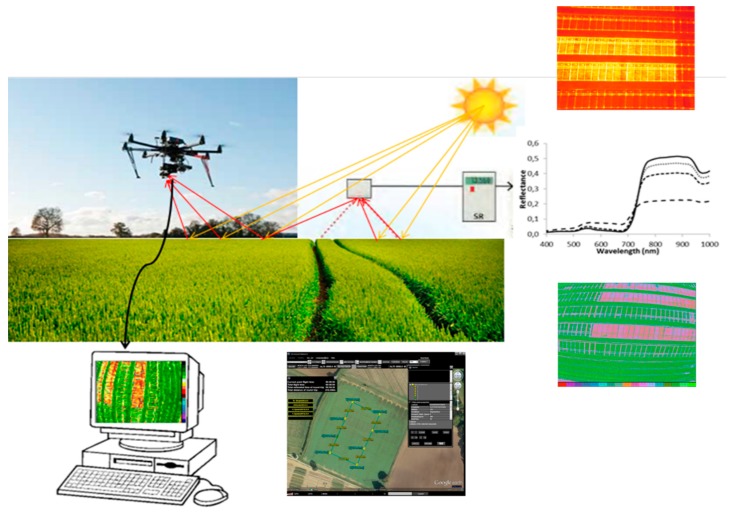
High throughput phenotyping using remote sensing. The production of sufficient food to meet increasing population demands while maintaining environmental sustainability is one of the greatest challenges of the twenty-first century. High throughput phenotyping (HTP) allows for precise monitoring of plant organs, individual plants, field plots, and full fields as required. These platforms include spectroradiometers, thermal sensors, Red–Green–Blue (RGB) imaging, etc. Due to the enormous volume and variety of imaging and remote sensing data generated, one of the main limiting factors is the management and interpretation of all this information. Further, connecting the different scales and platforms will be an important goal, and the key to creating useful phenotyping tools for the selection of genotypes with enhanced resource use efficiency under future stressful environments.

**Figure 4 plants-09-00088-f004:**
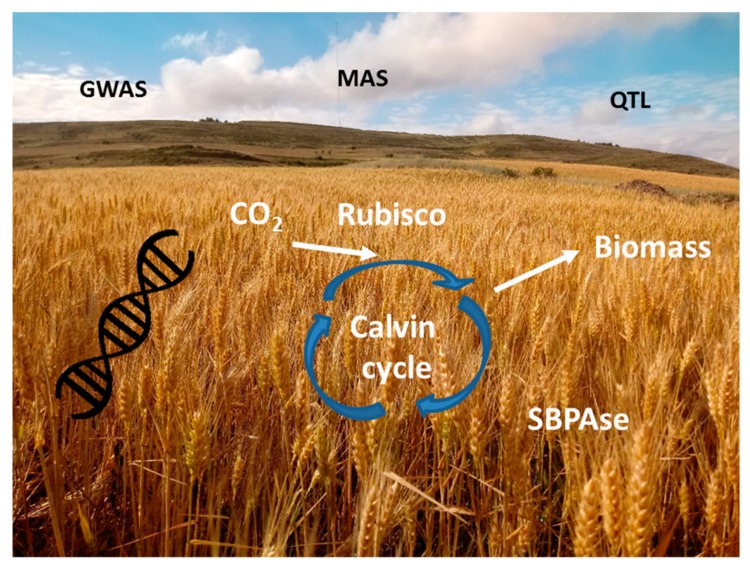
Photosynthesis-based multi-approach crop breeding. Plant photosynthesis, and therefore, growth is determined by multiple regulatory mechanisms. Previous studies have identified traits that may be targeted for modification to improve CO_2_ assimilation rates and yield. Alongside traditional breeding programs, modern techniques enable the development of synthetic biology and genetic engineering to develop crops that are better adapted to stress.

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
