# Peer review of "Photosynthetic Metabolism under Stressful Growth Conditions as a Bases for Crop Breeding and Yield Improvement"

_plants, 2020, doi:10.3390/plants9010088_

Round 1
Reviewer 1 Report
For centuries, food security has been a serious problem for humanity. Current climate change (manifested by the increase in surface temperature, intensification of the global hydrologic (water) cycle, changes in frequency and intensity of extreme atmospheric events (e.g., hurricanes, storms, heavy rainfalls) and hydrological extremes (e.g., floods and droughts) as well as the world population growth have given new impetus to this critical issue. Therefore, the topic of the article under review is very important and timely.
I suggest publishing paper after minor changes. There is a scientific and political consensus that the global surface temperatures will likely continue to rise, but this growth is characterised by some degree of uncertainty. However, an even greater degree of uncertainty is inherent in the Earth's water cycle, and in characteristics of extreme climatic events (their frequency, intensity and heterogeneity in spatial distribution). Since the water availability in critically important for food production, I suggest describing in more details how the climate change affects the hydrologic cycle and, particularly, precipitation, rainfall and floods. The attention should be also paid to projected geographical distribution of these features of the Earth's climate system. This will provide clearer picture of how climate change will affect food production in different parts of the world. Reading the paper, it is difficult to understand what the geographical area of interest is. Meanwhile, climate change will affect water availability in in different parts of the world in different ways, and therefore on food production.
Author Response
Since the water availability in critically important for food production, I suggest describing in more details how the climate change affects the hydrologic cycle and, particularly, precipitation, rainfall and floods. The attention should be also paid to projected geographical distribution of these features of the Earth's climate system. This will provide clearer picture of how climate change will affect food production in different parts of the world.
Following the recommendation made by the referee, in the Introduction section (page 2) reference to alterations in the hydrological cycle have been included.
Reading the paper, it is difficult to understand what the geographical area of interest is. Meanwhile, climate change will affect water availability in in different parts of the world in different ways, and therefore on food production.
As requested a more clear reference to different geographical distribution has been added to the manuscript (page 2).
Reviewer 2 Report
Dear Authors,
I have read the manuscript “Photosynthetic metabolism under stressful growth conditions” several times and, in my opinion, the paper could be of interest to the scientific community. However, I have some recommendations, which in my opinion will help the reader to understand better the information brought by the paper. I would recommend a major revision.
General Comments:
In my opinion, the title does not cover the whole range of the issues raised in the paper. It is limited rather to the informaton included in chapter 2 - Photosynthetic metabolism under a changing environment (with subchapters). Thus, I suggest modifying the title to be more adequate to the content. I also recommend adding an introduction section which allows the reader to know all aspects of the work and why they are discussed, adding aims of the study would be also beneficial. I think in some sense 1st Chapter could be partly used as an introduction to the paper. To make reading easier and allow the reader following the organization of the manuscript I suggest adding at the end of the introduction section short information regarding the order of raised issues, for example: We first address the photosynthetic metabolism under stress conditions, especially high temperature, water and nutrients scarcity. We then describe three major methods that have been used for photosynthesis-based breeding under the scenario of climate change. Finally, we discuss perspectives for crop breeding programs. In fact, the current chapter Perspectives summarizes the study, and no recommendations are given, therefore I suggest developing more deeply this section.
In my opinion, it would be beneficial to the manuscript if English native speaker revised the text.
Specific comments:
- Line 37: please remove the supplementary space between ‘threatening’ and ‘worldwide’
- Lines 53,55,60,63 etc.: please remove year in brackets after names.
- Line 103: please remove supplementary space between ‘plants’ and ‘can’.
- line 164: please remove supplementary space between ‘[77-78].’ and ‘Some’.
- line 177: please remove ‘es’.
Line 181: please remove supplementary space between ‘those’ and ‘related’.
-line 186: please remove supplementary ‘;’.
Line 191: please consider adding ‘still’ in: these techniques are still very expensive and time-consuming.
Line 251: Medicago trunculata – italic.
Line 251: Other enzymes.
Line 272: physicochemical properties.
Line 334-335: hard to read sentence.
Line 600, 620, 847: please remove the additional number.
Line 710: Keys, A.J.
Author Response
In my opinion, the title does not cover the whole range of the issues raised in the paper. It is limited rather to the informaton included in chapter 2 - Photosynthetic metabolism under a changing environment (with subchapters). Thus, I suggest modifying the title to be more adequate to the content.
Following the recommendation made, the title of the manuscript has been modified in order to make it more appropriate.
I also recommend adding an introduction section which allows the reader to know all aspects of the work and why they are discussed, adding aims of the study would be also beneficial. I think in some sense 1st Chapter could be partly used as an introduction to the paper. To make reading easier and allow the reader following the organization of the manuscript I suggest adding at the end of the introduction section short information regarding the order of raised issues, for example: We first address the photosynthetic metabolism under stress conditions, especially high temperature, water and nutrients scarcity. We then describe three major methods that have been used for photosynthesis-based breeding under the scenario of climate change. Finally, we discuss perspectives for crop breeding programs. In fact, the current chapter Perspectives summarizes the study, and no recommendations are given, therefore I suggest developing more deeply this section.
As requested, at the end of the Introduction section a short paragraph has been added aiming to summarize the main points addressed in the manuscript.
In my opinion, it would be beneficial to the manuscript if English native speaker revised the text.
The manuscript has been revised by an English Editing service.
Specific comments:
All the comments have been considered when writing the new version of the manuscript.
Round 2
Reviewer 2 Report
In the revised version of the manuscript, the Authors used my tips and addressed correctly the line-by-line suggestions. However, my general comments regarding "Perspectives" still require to be addressed. I recommend the article be accepted after deeper conclusions made from the current state of knowledge.
Specific comment: In my opinion, the previous version of the paragraph (Beginning with "Concurrent with...; lines 48-58) was more suitable to the content of the article than the current one (more developed).
Author Response
Following the recommendation made by the Referee 2, in the new version of the manuscript, we have extended the “Perspectives” section in order to more clearly remark the relevance of current (and near future) research on photosynthetic apparatus under different environmental conditions (page 14, lines 540-550).
Regarding the comment made by the Referee 2 on paragraph located between lines 48-58, we would like to mention that such modification was requested (and agreed) by the Referee 1.